# Radioprotective Role of Vitamins C and E against the Gamma Ray-Induced Damage to the Chemical Structure of Bovine Serum Albumin

**DOI:** 10.3390/antiox10121875

**Published:** 2021-11-24

**Authors:** Hajar Zarei, Mostean Bahreinipour, Yahya Sefidbakht, Shokouh Rezaei, Rouhollah Gheisari, Susan Kabudanian Ardestani, Vuk Uskoković, Hiroshi Watabe

**Affiliations:** 1Physics Department, Persian Gulf University, Bushehr 75169, Iran; gheisari@pgu.ac.ir; 2Cyclotron and Radioisotope Center (CYRIC), Tohoku University, Sendai 980-0845, Miyagi, Japan; 3Department of Energy Engineering and Physics, Faculty of Physics, Amirkabir University of Technology, Tehran 1591634311, Iran; mostean1398@gmail.com; 4Protein Research Center, Shahid Beheshti University, Tehran 1983969411, Iran; y_sefidbakht@sbu.ac.ir (Y.S.); shokouhRezaei956@gmail.com (S.R.); 5Nuclear Energy Research Center, Persian Gulf University, Bushehr 75616, Iran; 6Institute of Biochemistry and Biophysics (IBB), University of Tehran, Tehran 1417614411, Iran; ardestani@ut.ac.ir; 7TardigradeNano LLC., Irvine, CA 92604, USA; vuskokovic@sdsu.edu or; 8Department of Mechanical Engineering, San Diego State University, San Diego, CA 92182, USA

**Keywords:** protein, gamma radiation, radioprotection, vitamin C and E

## Abstract

Radioprotective effects of vitamin C and vitamin E as a water-soluble and a lipid-soluble agent, respectively, were investigated at the molecular level during the imposition of gamma radiation-induced structural changes to bovine serum albumin (BSA) at the therapeutic dose of 3 Gy. Secondary and tertiary structural changes of control and irradiated BSA samples were investigated using circular dichroism and fluorescence spectroscopy. The preirradiation tests showed nonspecific and reversible binding of vitamins C and E to BSA. Secondary and tertiary structures of irradiated BSA considerably changed in the absence of the vitamins. Upon irradiation, α-helices of BSA transitioned to beta motifs and random coils, and the fluorescence emission intensity decreased relative to nonirradiated BSA. In the presence of the vitamins C or E, however, the irradiated BSA was protected from these structural changes caused by reactive oxygen species (ROS). The two vitamins exhibited different patterns of attachment to the protein surface, as inspected by blind docking, and their mechanisms of protection were different. The hydrophilicity of vitamin C resulted in the predominant scavenging of ROS in the solvent, whereas hydrophobic vitamin E localized on the nonpolar patches of the BSA surface, where it did not only form a barrier for diffusing ROS but also encountered them as an antioxidant and neutralized them thanks to the moderate BSA binding constant. Very low concentrations of vitamins C or E (0.005 mg/mL) appear to be sufficient to prevent the oxidative damage of BSA.

## 1. Introduction

Gamma radiation is a form of ionizing radiation used extensively in medicine, for example, in nuclear diagnostic imaging or in the treatment of malign neoplasms [1]. Gamma rays are electromagnetic waves with frequency in excess of 10^19^ Hz and wavelengths lower than 100 pm, produced during the decay of the atomic nuclei of radioactive isotopes. Because of their ability to penetrate matter deeper than other, more particulate forms of radiation, such as alpha or beta particles, they are commonly used to interact with deep tissues to achieve various diagnostic or therapeutic goals. Many classical and advanced medical tools, such as emission tomography or gamma knife radiosurgery, make use of gamma radiation. As the application of methods based on ionization radiation, in general, is on the rise among medical professionals, more attention must be paid to its side effects and associated risks. For example, during radiation cancer therapy, noncancerous cells, such as endothelial cells, may be damaged, especially at high radiation doses, thus adversely affecting the therapeutic outcome [2].

Radiolysis of water by ionizing radiation is a major source of reactive oxygen species (ROS) in cells and tissues irradiated by gamma rays. Oxidative stress conveyed through the ROS activity causes deleterious effects on biomolecules, including protein oxidation, misfolding and aggregation, DNA damage and mutations, and lipid peroxidation [3,4]. The ROS can also induce changes to the secondary and tertiary structure of various proteins, including myoglobin, human and bovine serum albumins (HSA and BSA), and sunflower protein [5,6,7,8]. The ROS generated by radiation could also modify the primary structure of the proteins via degradation, cross-linking, and aggregation of polypeptide chains [5], which results in distortions of the secondary and tertiary structures [6]. The critical factor in many age-related diseases, particularly the various neurodegenerative diseases such as Alzheimer, Parkinson, or Huntington diseases, is the oxidation of proteins by the reactive products of radiation, which leads to aggregated or misfolded protein forms as molecular triggers of the given pathologies [7].

For this reason, the protection of protein molecules in the body is the primary concern when the imbalance between the ROS production and antioxidant activity occurs due to radiation exposure, as in radiotherapy. Overall, there is a great need to improve the radiation safety of patients undergoing radiotherapy through the use of appropriate radioprotectors.

Many prior studies assessed the potentials of various synthetic and natural compounds to act as radioprotection agents. To this date, only Amifostine has been approved by FDA for patients undergoing radiotherapy. However, the prescription of this drug is limited due to pronounced side effects [8,9]. The studies are, therefore, continuing to find and develop synthetic or natural compounds as radioprotectors with higher biocompatibility, nontoxicity, efficacy, and stability. Among them, vitamins C and E and their derivatives have attracted the attention of researchers, given that innumerable in vitro and in vivo studies have confirmed the ability of these two compounds to manage oxidative stress [3].

Vitamin C is the generic name for ascorbic acid. The chemical structure of this water-soluble vitamin composed of hydroxyl groups bound to a furan ring makes it a relatively suitable donor of electrons and protons. Therefore, this vitamin can be oxidized simultaneously, as it reduces ROS compounds, such as the superoxide anion radical. Competitive reactions of vitamin C with ROS cause the latter to be neutralized before they reach the critical compartments of the cell [10,11]. Vitamin E, in turn, is the generic name for all biologically active stereoisomeric compounds consisting of tocopherol and tocotrienol. These chemical groups are well-known for their potent antioxidant properties and the ensuing ability to scavenge ROS and free radicals. This ability is due to the hydroxyl group on the chroman ring, which is a strong donor of protons that reduce free radicals [12].

Radioprotection experiments in this study were carried on BSA as a model protein. BSA is not only one of the most significant blood carrier proteins, but it is also one of the candidate protein markers for radiation biodosimetry based on the high damage degree of the protein by gamma radiation [13]. In previous studies, the effects of different commonly used doses (3 and 5 Gy) of gamma radiation on the molecular structure, size distribution, and surface charge of BSA were investigated by several spectroscopic methods. The studies have revealed a significant effect of gamma rays on the secondary and the tertiary structure of the protein [14,15].

The analyses of changes made to the secondary and the tertiary structure of BSA by irradiation with and without the vitamins were conducted using circular dichroism (CD) and fluorescence spectroscopies. CD spectroscopy is a proven method for assessing the secondary structure of proteins based on the unique degrees of ellipticity at characteristic wavelengths of polarized light produced by different structural alignments of amide chromophores in an amino acid chain [16]. CD spectroscopy allows for the discernment of different secondary structural motifs in a protein based on the unique CD spectra generated by each. Fluorescence spectroscopy of proteins, in turn, is based on measuring the intrinsic fluorescence of optically active amino acid side chains, including those of Trp, Tyr, and Phe, upon appropriate optical excitation. This spectroscopy allows for the acquisition of information regarding the tertiary structure of proteins based on the intensity changes in fluorescence, which correlate with the 3-D localization of these three fluorophores [17].

To the best of our knowledge, there has been a persistent lack of studies investigating the antioxidant and radioprotective effect of vitamins C and E on macromolecules such as proteins [18], although there have been numerous studies performed in vivo [19,20,21]. Therefore, in this study, the radioprotection effect of vitamins C and E on 3 Gy dose gamma radiation-induced BSA structural changes were investigated to gain a better understanding of the potential for application of these common dietary ingredients in improving the safety of radiotherapy.

## 2. Materials and Methods

### 2.1. Reagents and Apparatus

Bovine serum albumin (BSA) (used without further purification), sodium chloride (NaCl), sodium hydroxide (NaOH), potassium dihydrogen phosphate (KH_2_PO_4_), potassium hydrogen phosphate (K_2_HPO_4_), and acetonitrile were purchased from Merck (Darmstadt, Germany). Vitamin E and vitamin C were purchased from Eastman (Eastman Kodak Company, Rochester, NY, USA) and Merck (Germany), respectively. The solutions were prepared in deionized double distilled water (Barnstead, Nano pure infinity, Dubuque, IA, USA), and all experiments were carried out at room temperature. Spectroscopic measurements were performed using a spectrofluorometer (Carry Eclipse, Varian, and Australia) and Circular Dichroism Spectrometer (Model 215, Aviv, New Jersey, NJ, USA). Each BSA sample was prepared in no less than triplicates and then scanned using CD and the fluorescence spectrophotometer. All experiments were conducted at room temperature.

### 2.2. Sample Preparation and Gamma Irradiation

First, a stock containing 0.8% and 0.1% of BSA and different concentrations of vitamins C or E were prepared in 10 mM PBS. Vitamin E is fat-soluble, and a small amount of acetone was used to dissolve it in the buffer. To prepare 0.1% stock solution of vitamin E, 1 mg of vitamin E was dissolved in 4 μL of acetone, and the final volume reached 1 mL with the addition of the buffer. Then, BSA samples with the final concentration of 0.4 mg/mL (0.04% *w/v*) were prepared in the absence and in the presence of vitamins C or E with 0.002, 0.005, 0.0075, 0.0125 and 0.02 mg/mL concentrations. The prepared BSA samples with and without vitamins C or E were irradiated at room temperature using ^60^Co gamma ray irradiator (Theratron 780-E, Canada) with a dose rate of 10 Gy/min. Irradiation experiments were performed at the Imam Khomeini Hospital (Tehran, Iran). In this treatment, BSA absorbed gamma radiation at total doses of 0.1, 0.5, 1, 2, and 3 Gy. The source skin distance (SSD) and the field of view (FOV) of irradiation were 80 cm and 25 cm × 25 cm, respectively. To achieve the electron balance in the wall of the glass vials containing the protein samples during irradiation, the vials were placed in a large water pool made of plexiglass (20 × 20 × 10 cm^3^). The plexiglass container was located in front of the ^60^Co source, at the center of the FOV.

### 2.3. Circular Dichroism (CD) and Fluorescence Spectroscopy

The far CD measurements of all nonirradiated BSA and irradiated BSA (IR-BSA) samples were performed in the 190–260 nm range and at the 200 nm/min scanning speed. All of the CD spectra were baseline-subtracted by using a spectrum of the solvent obtained under the same experimental conditions. Raw data were analyzed using the CDNN 2.1 software (Circular Dichroism analysis using Neural Networks). The fluorescence emission intensity of BSA and IR-BSA were recorded by excitation at 280 nm in the 300–440 nm range.

### 2.4. Molecular Docking

4F5S X-ray diffraction structure of BSA with 2.47 Å resolution retrieved from PDB (Protein Data Bank) was used for all simulation and docking studies [22]. Electrostatic properties of the protein were calculated using PDB2PQR, which solves the equations of continuum electrostatics for biological macromolecules [23]. CB-Dock was used to predict the possible binding modes in the absence of additional information about binding sites [24]. This algorithm utilizes a novel curvature-based protein cavity detection approach to predict binding sites followed by docking with Autodock Vina based on the calculated centers on the protein’s surface [24,25]. Ligplot (v.1.4.5) was applied to analyze protein-ligand contacts by representing the 2D ligand–protein interaction diagrams [26]. Protein surface properties were calculated using Protein Interfaces, Surfaces and Assemblies service (PISA) and Volume, Area, Dihedral Angle Reporter (VADAR) [27,28]. UCSF Chimera software (1.15) was used to represent the 3D models of the protein [29].

## 3. Results

### 3.1. Effect of Gamma Radiation at Therapeutic Doses on the Secondary and Tertiary Structure of BSA

To investigate changes caused by gamma rays to the structure of the BSA protein at the therapeutic doses of 0.1, 0.5, 1, 2, and 3 Gy, CD and fluorescence spectroscopy were used to study the secondary and the tertiary structure, respectively.

CD spectroscopic data obtained in the far-UV region and intrinsic fluorescence spectra of BSA and IR- BSA solutions irradiated with the therapeutic doses of gamma rays are shown in Figure 1A,B. The former spectra exhibit a signal characteristic of the α-helix structure with two negative bands in the far-UV region at 208 and 222 nm [30], originating from the n→π* transfer for the peptide bond in α-helix [31]. To gain a more quantitative structural insight, the percentage of the secondary structure elements in control BSA and IR-BSA for different therapeutic radiation doses was analyzed by the CDNN software.

BSA is a single-chain midsize protein composed of 583 residues and has a molecular weight of ~66.5 kDa. It is a water-soluble globular protein, and its main secondary structure motifs are helices, which form 67% of its structure [32,33]. The results displayed in Table 1 indicate that irradiation caused a 5.5–7% decrease in the content of α-helices in the protein, an increase of 2.5–3.5% in the content of random coils, and an increase of 1.5–3% in the content of β-structures. Therefore, irradiation can be said to have caused significant changes to the secondary structure of irradiated samples as compared to nonirradiated control samples. In addition, the α-to-β helix transfer in irradiated protein samples increased with the radiation dose. Irradiation at 3 Gy, correspondingly, displayed the highest α-to-β helix conversion and irradiations at 0.1 and 0.5 Gy the lowest. Still, the degree of helix conversion achieved with an increase of the radiation dose from 0.1 to 3 Gy was at around 1%, only a portion of the 9% change observed at the 0.1 Gy dose relative to the nonirradiated BSA. This suggests that the radiation-induced changes to the protein structure, if not the number of ROS produced, are only marginally higher at the highest tested dose of 3 Gy compared to the lowest dose of 0.1 Gy. Even one such relatively low radiation dose is sufficient to produce considerable changes to the protein structure, almost identical to those observed at 3 Gy.

The intrinsic fluorescence results indicate a significant reduction in the emission intensity of IR-BSA samples compared with the nonirradiated sample. Earlier, the irradiation of BSA at therapeutic doses caused tryptophan (Trp) residues buried in hydrophobic regions to relocate to a hydrophilic region and reduce fluorescence intensity [34]. This change in the location of tryptophan residues is directly related to the radiation dose. As a result, the environment around the Trp residues becomes progressively more hydrophilic with increasing the radiation dose. The maximum emission wavelength for all samples was approximately constant and detected at 345 nm.

### 3.2. Investigation of the Effect of Gamma Rays at the 3 Gy Dose on the Structure BSA in the Presence of Natural Protectors

#### 3.2.1. Radioprotection of the Secondary Structure of IR-BSA with Vitamin C

Before the radioprotection properties of vitamin C and E can be assessed, it is necessary to consider their direct effects on the conformational changes of BSA in the absence of gamma radiation. Accordingly, changes to the secondary and tertiary structures of BSA and IR-BSA in the presence of different concentrations of vitamin C and vitamin E were studied first and compared using far-UV CD and fluorescence spectroscopy.

The CD spectra of BSA and IR-BSA in the far-UV region at the radiation dose of 3 Gy in the presence of different concentrations of vitamin C are shown in Figure 2A,B, respectively. The results of the secondary structure content analysis are summarized statistically in Figure 3 for better review and comparison.

The α-helix content of BSA decreased with increasing the concentration of vitamin C (up to 0.008 mg/mL) and then reached an almost stable state. The secondary structure changes of BSA at low concentrations of vitamin C (≤0.005 mL), on the other hand, were negligible.

To evaluate the structural difference in BSA and IR-BSA in the presence of different concentrations of vitamin C, an unpaired t-test was performed using SPSS version 16(IBM). Two separate t-tests were performed to compare, (1) BSA in the absence of vitamin and BSA in the presence of vitamin, and (2) BSA in the absence of vitamin and IR-BSA in the presence of vitamin. We set a *p*-value of less than 0.05 (*) as statistically significant. As can be seen in Figure 3, the secondary structure content of IR-BSA is most similar to native BSA at low (≤0.002 mg/mL) and high (≥0.012 mg/mL) vitamin C concentrations. Vitamin C and ROS are two agents that can cause conformational changes to BSA as quenchers in the irradiated medium. At low concentrations of vitamin C, the contribution of non-neutralized free radicals—including ROS and the ascorbyl radical—on the protein structure change is greater than that of vitamin C, and the opposite applies at higher vitamin C concentrations. In other words, the damaging effects of ROS expectedly decline by increasing the concentration of vitamin C as the scavenging agent. The α-helix content of IR-BSA is higher than that of BSA at vitamin C concentrations >0.008 mg/mL, but still somewhat lesser than that of the native BSA.

#### 3.2.2. Radioprotection of the Tertiary Structure of IR-BSA with Vitamin C

Figure 4A,B shows the fluorescence spectra of BSA and IR-BSA in the presence of different concentrations of vitamin C, while Figure 5 shows maximal intensities of these spectra. The fluorescence intensity of BSA decreased with increasing the vitamin C concentration, indicating its quenching effect on the protein. This effect is due to increased exposure of hydrophobic areas (and subsequently fluorescing Trp, Phe, and Tyr residues buried in the hydrophobic core) in hydrophilic regions due to partial unfolding of BSA, which appears to be directly related to the concentration of the vitamin [17,34,35]. From these data, it can be concluded that vitamin C has a high binding affinity for BSA as a ligand.

The fluorescence intensity of IR-BSA also declined by increasing the vitamin C concentration (Figure 4B and Figure 5). IR-BSA exhibited a higher fluorescence emission intensity than BSA at comparative concentrations of vitamin C. This indicates that the direct interaction between vitamin C and ROS reduces the direct interaction between BSA and vitamin C. These results also indicate that the quenching effect of vitamin C on BSA at high concentrations is more dominant than the quenching effect caused by ROS species.

#### 3.2.3. Radioprotection of the Secondary Structure of IR-BSA with Vitamin E

Figure 6A,B shows the CD spectra of BSA and IR-BSA in the far-UV region at the radiation dose of 3 Gy in the presence of different concentrations of vitamin E. For a better review, the percentage of secondary structure elements extracted from Figure 6 is summarized in Figure 7.

The results confirm that the presence of vitamin E, regardless of the concentration, had minimal effect on the secondary structure of both BSA and IR-BSA, except for BSA at 0.002 mg/mL vitamin E. Therefore, vitamin E can be said to have produced negligible changes to the secondary structure of BSA. The low concentration exception may be because the hydrophobic affinity among the hydrocarbon tail of vitamin E molecules increases with concentration. As a result, the penetration of such vitamin aggregates into the BSA protein structure decreases as compared to the low concentration scenario, which in turn reduces secondary structure disturbances.

The results also reveal that the amount of α-helices in IR-BSA in the presence of vitamin E increases (6–7%) compared with IR-BSA in the absence of vitamin E. This applies to all vitamin E concentrations, which yield values similar to the content present in native BSA, indicating that even a low vitamin E concentration (0.002 mg/mL) can provide good radiation protection and prevent structural changes in BSA.

#### 3.2.4. Radioprotection of the Tertiary Structure of IR-BSA with Vitamin E

The intrinsic fluorescence spectra of BSA and IR-BSA in the presence of various concentrations of vitamin E are shown in Figure 8A,B, respectively, while Figure 9 shows maximal intensities of these spectra. The maximum emission wavelength for all the samples was approximately constant and detectable at 345 nm.

The fluorescence intensity of BSA decreased with increasing the concentration of vitamin E, indicating its quenching effect on the protein. Mechanistically, the presence of vitamin E caused changes in the microenvironment around Trp, Phe, and Tyr residues, exposing them to the hydrophilic solvent [17,34,35]. On the other hand, no significant difference between the emission fluorescence intensity of IR-BSA samples containing different vitamin E concentrations was generally observed. Their emission intensities were either somewhat greater or similar than those of IR-BSA in the absence of vitamin E.

According to Figure 8, IR-BSA samples containing vitamin E exhibited higher fluorescence intensities than nonirradiated BSA samples containing comparative concentrations of vitamin E, which indicates that the areas around Trp, Phe, and Tyr residues in BSA are more hydrophobic than those in IR-BSA. The difference between the maximum emission intensities of BSA and IR-BSA increased with increasing the vitamin E concentration. Based on these data, it can be inferred that vitamin E is an excellent radiation protector, capable of maintaining the tertiary structure of BSA and preventing the destructive effects of gamma radiation.

### 3.3. Fluorescence Quenching Mechanism and the Binding of Vitamins C and E to BSA

To further investigate the quenching mechanism of BSA and IR-BSA, as induced by vitamin C or E, the fluorescence data were analyzed with the Stern–Volmer equation [36,37]:F_0_/F = 1 + K_q_τ_0_ [Q] = 1 + K_SV_(1)
where F_0_ and F are steady-state fluorescence intensities in the absence and in the presence of the quencher, respectively, K_SV_ is the Stern–Volmer quenching constant, [Q] is the concentration of the quencher (i.e., vitamins), K_q_ is the bimolecular quenching rate constant, and τ_0_ is the average lifetime of the fluorophore in the excited state, with its value usually being 10^−8^ s for a biological macromolecule. The linearity of the F_0_/F versus [Q] plots for both BSA and IR-BSA is shown in Figure 10, whereas the estimated values of K_SV_ and K_q_ at room temperature are shown in Table 2. Two main points can be extracted from these data: (1) the quenching constant of vitamin C is larger than that of vitamin E for both BSA and IR-BSA, and (2) the quenching constants for both vitamins are lower for IR-BSA than for BSA. The quenching constant of vitamin C is about 1.7 and 13 times that of vitamin E for BSA and IR-BSA, respectively. In addition, the K_q_ for all BSA-vitamin C/E and IR-BSA–vitamin C/E combinations shown in Table 2 is larger than 2.0 × 10^10^ mol·L^−1^, which is the maximum diffusion collision quenching rate constant considering various quenchers of biopolymers. This reveals that the quenching fluorescence mechanism most probably follows a static quenching process rather than a dynamic one. The fluorescence data were further examined using the modified Stern–Volmer equation [36,37]:F_0_/∆F = [1/f_a_ K_a_][1/[Q]] + 1/f_a_(2)
where f_a_ is the fraction of the initial fluorescence accessible to the quencher; K_a_ is the effective Stern–Volmer quenching constant of the accessible fraction, and [Q] is the concentration of quencher. The F_0_/∆F (F_0_/F_0_ − F) versus 1/[Q] dependence is displayed in Figure 11. Both Stern–Volmer equations shown a linearly increasing trend from all vitamin-BSA combinations except for IR-BSA with vitamin E, which shows a partial (Figure 11) or complete (Figure 10) plateau. Therefore, the K_a_ value for the IR-BSA–vitamin E complex is not reportable. The results obtained using Equation (2) also reiterate that the quenching mechanism is based on static quenching.

The number of binding sites (n) and the binding constant (K) for the quenching interaction of vitamins C or E with BSA and IR-BSA can be calculated using Equation (3) [36,37]:log[F_0_ − F/F] = logK + n log[Q](3)
where F_0_, F, and [Q] are the same parameters as those in Equation (1). Equation (3) indicates the equilibrium between the free and the bound molecules. A plot of log [(F_0_ − F)/F] versus log [Q] gives a straight line (Figure 12), whose slope equals n and the intercept on the *Y*-axis equals log K. The values of K and n at room temperature are listed in Table 2. The values of n are approximately equal to 1 for BSA-vitamin C/E complexes and indicate the existence of a single binding site on BSA for both vitamins. In the case of vitamin C, the value of n does not change significantly depending on whether BSA is irradiated or not. However, for the IR-BSA–vitamin E complex, the value of n is reduced dramatically, by order of magnitude, down to only 0.1. The binding constant (K) for all BSA–vitamin combinations except IR-BSA–vitamin E are in the 1–150 × 10^4^ range, confirming the reversible binding and moderate affinity of vitamins to BSA [36]. Such a low value of n and K for the IR-BSA–vitamin E complex reveals the vitamin E dissociation from BSA in the presence of ROS. The binding constant of vitamin C is higher than that of vitamin E due to its hydrophilic properties and smaller size, which allow it to interact freely with BSA. Although the binding constant of the IR-BSA–vitamin C complex is greater than that of the BSA–vitamin E complex, it shows lower K_SV_ and K_a_.

### 3.4. Molecular Docking Analysis

Molecular docking simulations were performed to derive the possible binding sites for vitamins C and E on the surface of BSA and are illustrated in Figure 13. Additionally, 2D interaction plots for the lowest free energies of binding are represented in Figure 14. As can be inferred from the simulations, both vitamins can attach to various parts of the protein. However, lipophilic vitamin E is predominantly surrounded by hydrophobic side chains, with the polar chroman head being observable in the vicinity of the positively charged patch near Trp 213. Vitamin E binding is primarily based on van der Waals interactions, as can be seen in the C-terminal binding pocket, where multiple hydrophobic residues can be involved, including Phe 550, Val 546, 551, 546 and 556, Leu 528, Met 547, and Ala 527 (Figure 14B). Hydrophilic vitamin C, in contrast, generally interacts with polar and charged residues, namely Tyr 149, 156, Ser 191, and Arg 198 (Figure 14A).

Previous studies have found that among the amino acids, Tyr, Cys, Met, and Trp are particularly sensitive to ROS [38,39]. Since these sensitive residues are distributed on the surface of BSA, preference for different binding sites on the protein surface could be the critical determining factor for the antioxidant activity of the two vitamins. Movie 1 shows the distribution of possible binding sites involving the disulfide bonds and Trp and Met residues. Based on these results, the presence of the vitamins as protective molecular agents could lead to the following mechanisms of protection:The chemical effect, in which case vitamins scavenge radicals and thus prevent them from interacting with the protein.The steric effect, in which case the interaction of vitamins with BSA, causes steric alterations at or around the binding site, thereby inhibiting radicals from interacting with the protein. This scenario is important when the binding pocket is near the sensitive amino acids.The colloidal effect, in which case the presence of vitamins stabilizes the protein dispersion behavior and prevents aggregation.

The free energy of binding for vitamins C and E is predicted to be −10.7 and −7.0 kcal/mol. Here, vitamin E exhibits a higher affinity for hydrophobic patches of BSA, while smaller and water-soluble vitamin C prefers polar environments and gravitates around smaller binding pockets on the BSA surface. This might be the reason for the rapid fluorescence quenching effect exhibited by vitamin C. Therefore, while vitamin C floats in the solvent around the protein, vitamin E attaches to the hydrophobic parts of the BSA surface. Vitamin E binding always involves the hydrocarbon chains of multiple residues and depends on an accessible hydrophobic area. Due to hydrophobic interaction, this binding is stable in the close vicinity of the protein surface. In fact, the formation of multiple hydrophobic interactions involving the phytyl side chain makes this attachment notably stable.

The computational analysis comparing BSA with HSA showed that the human and the bovine have sequence identity (75.5%) and similarity (87.5%). However, these two homolog proteins have very common surface properties and the same isoelectric point of 5.4. The results of the comparative BSA and HSA surface analysis, taking into account the accessible surface area (ASA), are shown in Figure 15. HSA and BSA possess not only a nearly identical secondary structure content but also highly similar surface properties in terms of the hydrophobic/hydrophilic ratio. Hence, since the amount of vitamin E binding can be directly related to the inhibition of protein oxidation, it is likely that similar behavior as that evidenced on BSA would apply to HSA as well.

## 4. Discussion

Oxidative damage in proteins correlates with aging, environmental factors, and stress and could rapidly affect cell viability by diminishing the function of enzymes and other proteins [39,40]. These phenomena start with the oxidation of specific amino acids and are followed by protein degradation at the point when the protein can no longer compensate for the damage by altering its unique chemical and topological characteristics. Here, Cys, Met, Trp, and Tyr residues are believed to be most sensitive to ROS, even at low ROS concentrations resulting from exposure to low-dose radiation [38,39]. Generally speaking, the degree of oxidation for specific fragments of the polypeptide chain is directly related to the accessibility (based on RMSD) and the number of hydrogen bonds [41].

Previous studies indicated that degradation and aggregation of proteins were negligible at low radiation doses, but irreversible effects on the secondary and the tertiary structure of proteins were significant [6,14,15]. Thus, measurements in this study were focused on the secondary and tertiary structures of BSA, given that the damage imposed at that structural level precedes a more obvious deterioration of the protein structure and function. Therefore, investigation of the radiation effects on the primary structure of BSA, as with SDS-PAGE or chromatography, was ignored. Vitamins C and E are two types of natural molecules that have been chosen to investigate their protective effect on the BSA structure against gamma radiation delivered at the dose of 3 Gy. The radioprotective effect of these two vitamins against the protein damage induced by gamma radiation was investigated specifically as an initiatory stage of pathogenesis. Although the protective effect against oxidation-induced structural alterations evidenced in this study is likely to be relevant for the prevention of specific disease states associated with radiation exposure, this effect can also be viewed in a broader context. In what follows, we will discuss the radioprotection properties of these two vitamins first sequentially and then in comparison with one another and with a nano-protector of choice.

**Vitamin C:** The difference between the secondary structure contents of BSA and IR-BSA at low concentrations of vitamin C (less than 0.008 mg/mL) is mainly related to the presence of non-neutralized ROS due to the insufficient amount of vitamin C required to scavenge them. Meanwhile, the direct effect of vitamin C on BSA at low concentrations of the vitamin is negligible, considering almost the same secondary structure of BSA in the absence and in the presence of vitamin C in this concentration range. The structural difference of the IR-BSA protein molecules in the presence of vitamin C compared to native BSA stems primarily from their mutual binding, which remains intact in the presence of the ROS.

The α-helix content of IR-BSA at the vitamin C concentration higher than 0.008 mg/mL is lesser than that of the native BSA, but it is higher than that of the native BSA in the presence of vitamin C. The first of these two effects is due to the interaction of ascorbate or dehydroascorbate with BSA, alongside the definite scavenging of ROS species by vitamin C in the solution. The latter effect, on the other hand, may be related to the lesser net effect of ascorbate and dehydroascorbate than to ascorbate alone due to the conversion of O-H groups to =O in dehydroascorbate.

The free radical form of vitamin C, a.k.a. the ascorbyl radical, is generated by donating an electron to a ROS such as the OH radical, a main reactive species scavenged after irradiation. Two pairs of ascorbyl radicals in the solution react rapidly to convert, albeit disproportionately, to ascorbate and dehydroascorbate, which eventually convert to 2, 3-diketo-1-gulonic acid. The main form of ascorbic acid (AcsH2) becomes deprotonized (AscH-) at neutral pH of 7 [42].

Although the interaction of the highly reactive ascorbyl radical with BSA is possible, the radical form of vitamin C should not be particularly challenging for the structure of BSA because the ascorbyl radical rapidly converts to the nonradical form. The effect, however, is expected to be more prominent at higher vitamin concentrations and radiation doses when the production of the ascorbyl radical in water due to radiolysis is more pronounced. The relative molar concentrations of vitamin C to BSA corresponding to 0.002, 0.005, 0.008, 0.0125, 0.02, and 0.025 mg/mL vitamin C are 1.9, 4.7, 7, 12, 19, and 24, respectively, and the effects of the ascorbyl radical are expected to be more elicited at these higher relative concentrations.

**Vitamin E:** Contrary to Vitamin C, which is hydrophilic (logP = −1.8), vitamin E (α-tocopherol) is hydrophobic (logP = 12.2), and its structure is composed of two parts: the chromane head and the phythyl tail, where the OH head group is responsible for interaction with the radicals and the antioxidant activity. In accord with the earlier studies [35,43], our work has confirmed that the interaction between vitamin E and BSA proceeds mainly via hydrophobic forces. Therefore, vitamin E has the natural propensity to localize to the BSA protein surface through hydrophobic interaction with the hydrophobic patches on the protein surface wherefrom it may exert its protective effect.

The CD results unequivocally demonstrate that the secondary structure of IR-BSA is protected by vitamin E, even at very low vitamin doses. Under this condition, the generated ROS diffuse through the water toward the BSA–vitamin E complex to encounter them. The interaction of ROS with vitamin E here is more probable and greater than that with the BSA protein because of the higher concentration of vitamin E and the shell effect around BSA, which leads to its physical shielding [44]. Namely, the molar concentration ratios of vitamin E to BSA corresponding to 0.002, 0.005, 0.008, 0.0125, 0.02, and 0.025 mg/mL vitamin E are 1, 2, 3, 5, and 10, respectively. Therefore, the diffusing ROS encounter vitamin E in this outer protective layer and convert it to α-tocopherol radical by altering the OH group of the chroman head. This radical is fairly stable due to the unpaired electron of oxygen delocalized in the aromatic ring and is not reactive enough to initiate protein peroxidation itself. Rather, it tends to recombine with another radical to convert to a nonradical product [45]. Most likely, α-tocopherol radicals attack one another, producing α-tocopherol dimers and trimers, but also 4a, 5- epoxy-8a-hydroperoxy α-tocopherol. While α-tocopherol dimers and trimers may cause spatial hindering that prohibits their mutual interaction and the interaction of diffusing ROS with BSA, α-tocopherol dimers and trimers have a relatively high reaction rate constant (in the order of 10^3^) [46]. All in all, the presence of vitamin E in IR-BSA samples effectively neutralizes ROS and prevents its binding to BSA through the inherent antioxidant activity and the production of new molecular forms.

**The comparison between vitamins C and E:** Generally, vitamins C and E act as ligands for BSA with moderate binding constants. Hydrogen bonds and hydrophobic interactions are the main driving forces for the binding of vitamins C and E, respectively, to BSA.

Vitamin C is significantly more polar and smaller than vitamin E. Therefore, it can form hydrogen bonds around different regions of the protein and reduce the solvation of the protein by lowering the number of hydrogen bonds formed between the protein and the water molecules solvating it. On the other hand, due to its relatively small size, it is more likely to reach groves on the protein surface that are not well fitted for bulkier ligands. This may explain the broader range of fluorescence quenching accompanied by slight secondary structure changes observed in BSA stabilized by vitamin C.

The CD results showed that the secondary structure contents of both BSA and IR-BSA are preserved in the presence of vitamin E, indicating the surface hydrophobic interactions could not cause substantial changes to BSA. The results also indicate that the oxidized molecular forms of vitamins C and E in IR-BSA samples, i.e., the dehydroascorbate and α-tocopherol dimers and trimer, respectively, have a lower quenching effect on BSA as compared to nonirradiated samples. Together, along with the different antioxidant mechanisms of vitamins C and E, the difference in the distribution of these vitamins on the BSA surface and the hydrophobic/hydrophilic ratio of the protein seem to be very important for a proper study of the radioprotective effects of vitamins. In any case, a very low concentration of both vitamins C and E, in the order of 0.005 mg/mL, is sufficient to strongly scavenge the ROS produced by ionizing radiation.

Whereas the scavenging activity of vitamin C is primarily exerted in the solvent, the suppression of ROS by vitamin E stems from its ability to bind to BSA. This is in agreement with the physiological fact that while vitamin C does not bind to serum proteins and circulates through the blood in the form of a free ascorbate anion [47], vitamin E is transported by a specific vitamin D binding protein, a member of the albuminoid family of proteins [48]. One question to consider at this point is whether additional drugs, drug carriers, or biomolecules would compete for the BSA surface with the vitamins and reduce their antioxidant efficacy. In such scenarios, exerting the mechanism of action in a diffused state, as with vitamin C, may prove to be an advantage. On the other hand, such synergetic activities may prompt the dehydrogenated vitamin C and vitamin E dimers and trimers mentioned above to lose their antioxidant activities. The formation of radical forms of vitamins is coupled to recycler pathways to convert them via chemical reduction to the nonradical native form in vivo [45,49]. Such mechanism intrinsically lacks in an in vitro setting and may drastically affect the antioxidant activities of vitamins.

It is worth mentioning that vitamins C and E exhibit the same effect as some common ROS scavengers. An in vitro study on irradiated human blood lymphocytes, for example, indicated that all common ROS scavengers including N-acetylcysteine (NAC), ß-carotene, selenium, vitamin E, vitamin C, and Q10 led to a significant reduction of double-strand breaks [50]. In another study, the radiation protection capacity of vitamins C and E with respect to human blood lymphocytes led to a significant reduction in X-ray-induced chromosomal damage, interestingly greater than that achieved by Amifostine, a drug characterized by high toxicity [51]. In addition, an in vivo study also showed that the radio-protective effect of vitamin C and E is equal to that of NAC in terms of preventing hepatoxic effects of amiodarone drug [52]. Compared to many of these drugs, the advantage of vitamins C and E lies in their nontoxic nature, alongside general dietary benefits.

**Comparison between vitamins C and E and a nano-radioprotector:** In our previous study, the radioprotective effect of synthesized ceria nanoparticles (CNPs) and magnetic flower-like iron oxide microparticles (FIOMPs) on irradiated BSA was investigated [14]. Some differences and similarities between these NPs and MPs with respect to the radioprotection of vitamins C and E will be discussed here.

First, the binding strength of the radioprotector to BSA is the key factor in determining its availability to disassociate reversibly from the protein and scavenge the ROS effectively. The K values indicated that the binding of vitamins C and E to BSA (in the order of 10^4^) and of NPs/MPs to BSA (less than 10^3^) is moderate and very weak, respectively. Both systems, especially the NPs/MPs, are bound reversibly to BSA, with the caveat that most NPs/MPs are unbound to BSA. Although the low affinity of the radioprotector to its target can be an advantage because it produces lesser side effects on the conformational changes of the protein, it cannot be an important factor in the case that it reversibly detaches to play its antioxidant and ROS scavenging roles. The latter role is clearly displayed by both vitamins C and E when the BSA sample is exposed to gamma radiation, and the vitamins neutralize the ROS in the solvent and radical amino acids on the protein, respectively, with their affinity to the protein per se being reduced. In other words, the reversible binding of vitamins C and E to BSA is not only important in preserving the ability of albumin to act as a plasma carrier protein open for other targets but also in achieving this ability to detach from the protected protein to interact effectively with the radicals.

Two general categories of antioxidants are found in the body: nonenzymatic ones, such as vitamin C, vitamin E, minerals, etc., and enzymatic ones, such as catalase (CAT), superoxide dismutase (SOD), glutathione peroxidase, etc. [48,53]. The CNPs/FIOMPs radioprotectors are an enzymatic type due to enzyme mimetic activities, including CAT, SOD, and peroxidase [45,53]. The main difference between them is that the radical forms of nonenzymatic antioxidants need to convert back to the molecularly stable form by the recycler system. Such recycler enzymes or other biomolecules are present naturally in vivo but are challenging to implement in an in vitro study, such as the one conducted here. Therefore, there is a pending concern about the vitamin radical attack on amino acids of the BSA protein, whereas no such concern applies to NPs/MPs radioprotectors. This concern for both vitamins C and E can be considered negligible for the aforementioned vitamin to BSA concentration ratios, although some attack of the radical vitamin forms on the BSA protein may still be present. It is interesting that based on this argumentation, the in vitro radioprotection of inorganic nano-radioprotectors can be more similar to the in vivo scenario than the in vitro radioprotection of organic molecules namely vitamins C and E. Nevertheless, the clear and precise idea about the difference between the mechanism of action of these two antioxidant agents is not easy to infer due to the complexity of the mechanisms involved, particularly in vivo.

Taken together, the natural and the synthetic radioprotectors discussed both show good radioprotection properties according to in vitro research presented here and elsewhere [14].

## 5. Conclusions and Future Directions

In this study, the radioprotective effect of vitamins C and E was investigated with respect to the gamma radiation-induced structural changes to BSA as a model protein at the therapeutic dose of 3 Gy. Various prior cellular and in vivo reports on the potential of vitamins C and E for radioprotection motivated our group to investigate this potential on the molecular level.

The preirradiation analyses carried out using CD and fluorescence spectroscopies showed nonspecific and reversible binding of vitamins C and E to BSA. Spectroscopic measurements further showed that gamma irradiation of BSA in the absence of vitamins results in changes of protein conformation accompanied by a decrease in the number of alpha-helices, an increase in the amount of complementary secondary structures, and a decrease in the fluorescence emission. In the presence of either of the two vitamins, irradiated BSA was protected from the structural changes induced by radiation. The reversible binding of vitamins C and E to BSA ensures that the vitamins dissociate easily and scavenge ROS, after which their radical forms convert to the native ones in vivo via a related biomolecule or enzyme, and BSA can transport them to the target tissue. At reasonable concentrations, vitamins C and E are not only nontoxic but are also predisposed for optimal antioxidant activity. The results of this study may confirm that the prescription of these vitamins after radiotherapy is a viable option, especially because these molecules exhibit no side effects, unlike other common drugs.

It is also expected that the results of this study on serum albumin as a primary and abundant blood protein can be used as a biomarker for the design of a novel biological dosimeter. Finally, regarding the complementary advantages of natural radioprotectors such as vitamins C and E and the synthetic ones such as the nano-material radioprotectors discussed earlier in the text, including low toxicity for the former and higher bioavailability due to resistance to digestion and potential for surface functionalization to yield enhanced target-specificity for the latter, a radioprotection system based on the synergic effect of these two radioprotectors can be proposed.

## Figures and Tables

**Figure 1 antioxidants-10-01875-f001:**
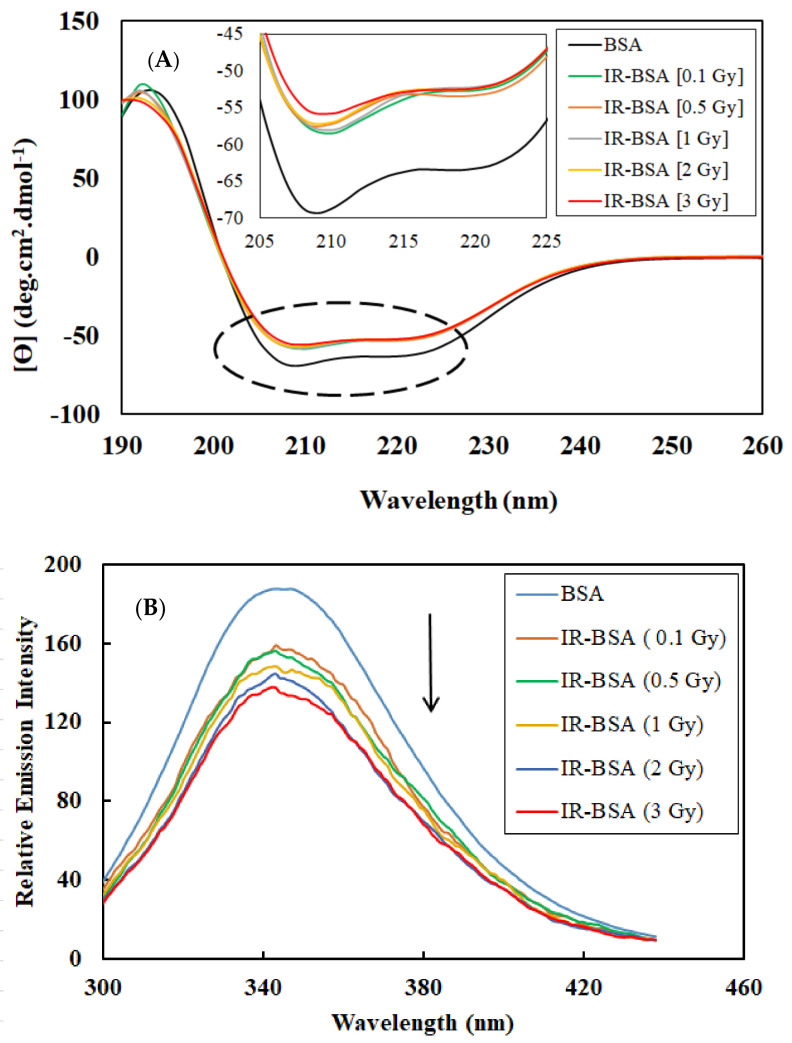
CD spectra in the far-UV region (**A**) and intrinsic fluorescent emission spectra after excitation by 280 nm wavelength (**B**) of BSA and IR-BSA solutions at therapeutic radiation doses [0.1–3 Gy]. All spectra were obtained on solutions containing the protein concentration of 0.4 mg/mL and prepared in a 10 mM phosphate buffer at pH = 7 at room temperature.

**Figure 2 antioxidants-10-01875-f002:**
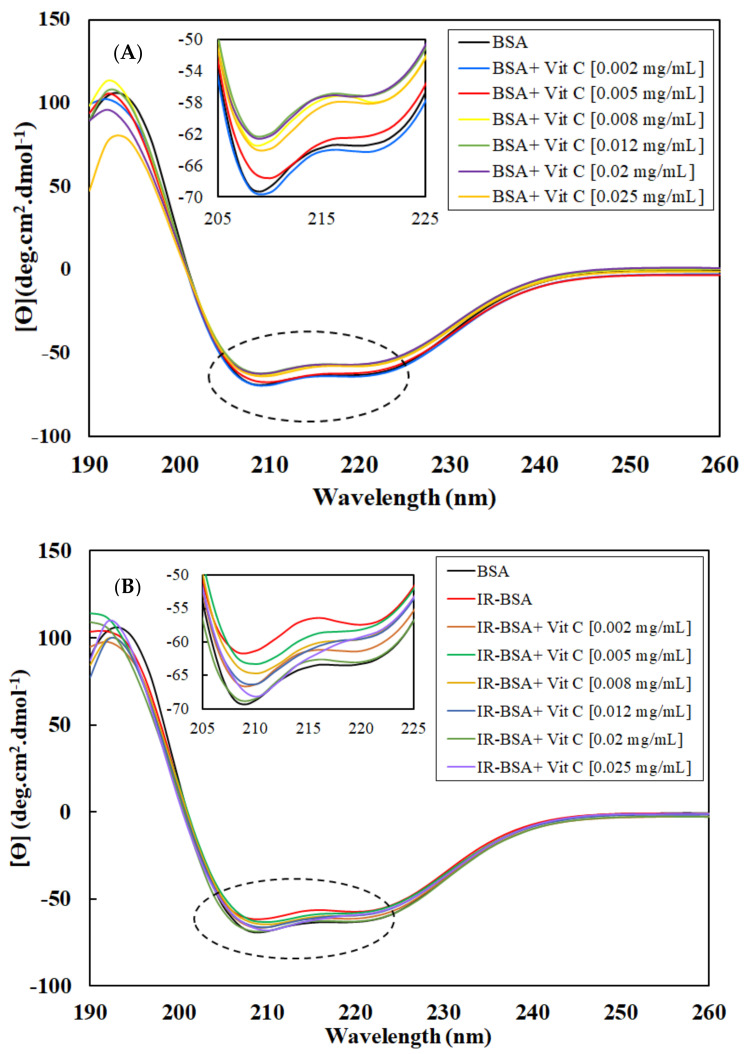
CD spectra in the far-UV region: (**A**) BSA and (**B**) IR-BSA (3 Gy) in the presence of different concentrations of vitamin C. All samples analyzed contained BSA at the concentration of 0.4 mL/mg in 10 mM phosphate buffer (pH = 7) at room temperature.

**Figure 3 antioxidants-10-01875-f003:**
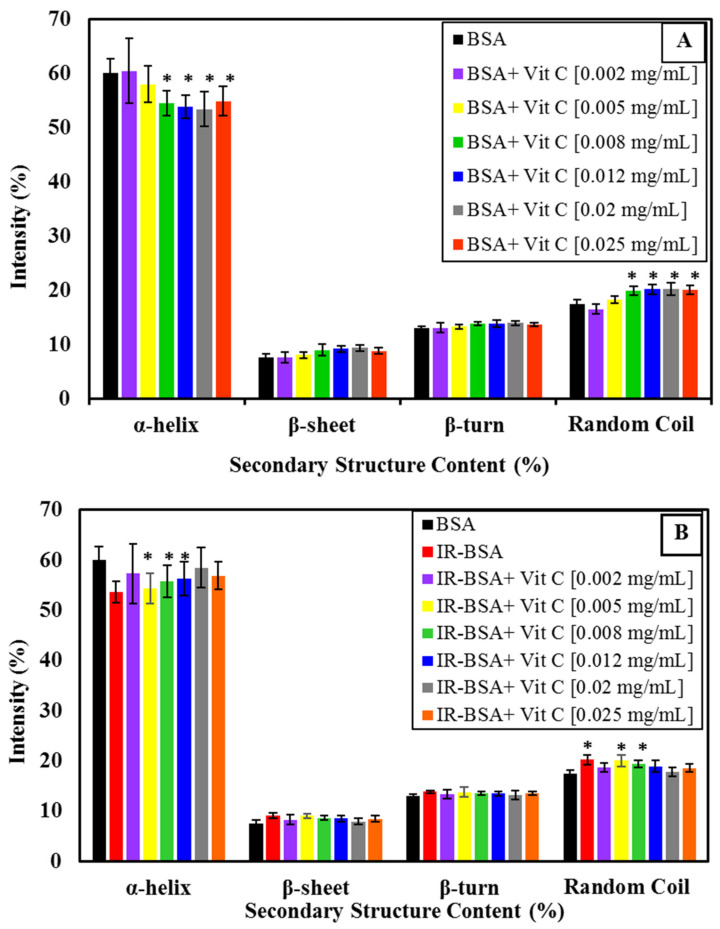
Percentage of secondary structural motifs of BSA (**A**) and IR-BSA (**B**) in the presence of different concentrations of vitamin C obtained from the analysis of the CD spectra using the deconvolution software CDNN2.1. The error bars represent the standard deviation of five measurements. The * sign indicates a *p*-value < 0.05 and a statistically significant difference.

**Figure 4 antioxidants-10-01875-f004:**
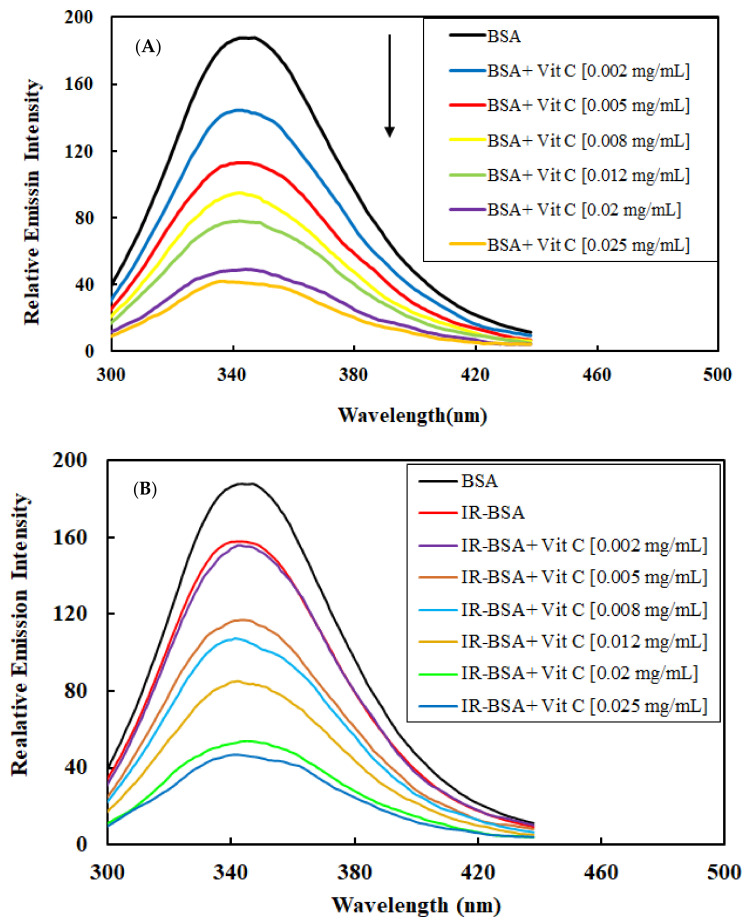
Intrinsic fluorescence emission spectra of BSA (**A**) and IR-BSA (**B**) in the presence of different concentrations of vitamin C. The solutions containing 0.4 mg/mL BSA in 10 mM PBS (pH 7.0) were excited at 280 nm, and the emission spectra were recorded in the 300–440 nm range.

**Figure 5 antioxidants-10-01875-f005:**
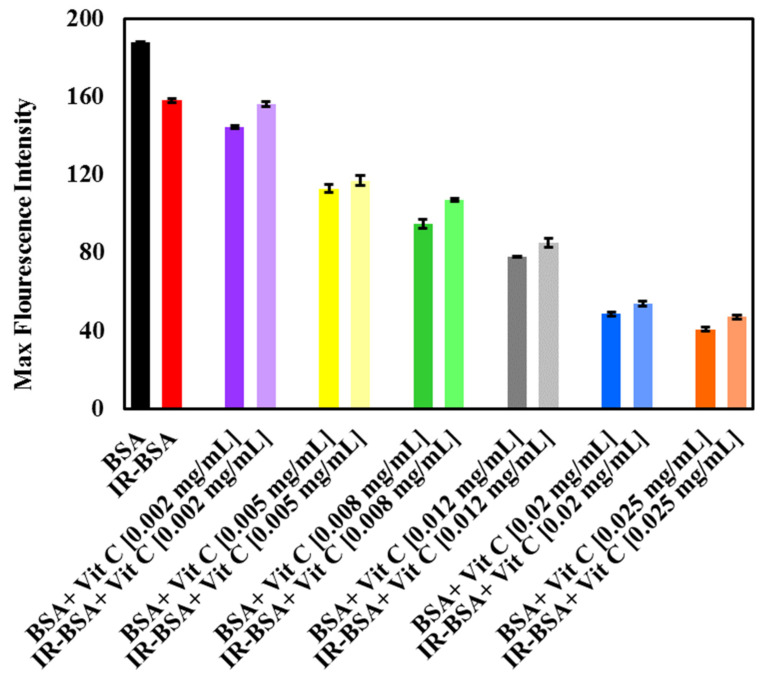
The maximum fluorescence emission of BSA and IR-BSA treated with different concentrations of vitamin C (Vit C). The experimental conditions were the same as mentioned in Figure 4.

**Figure 6 antioxidants-10-01875-f006:**
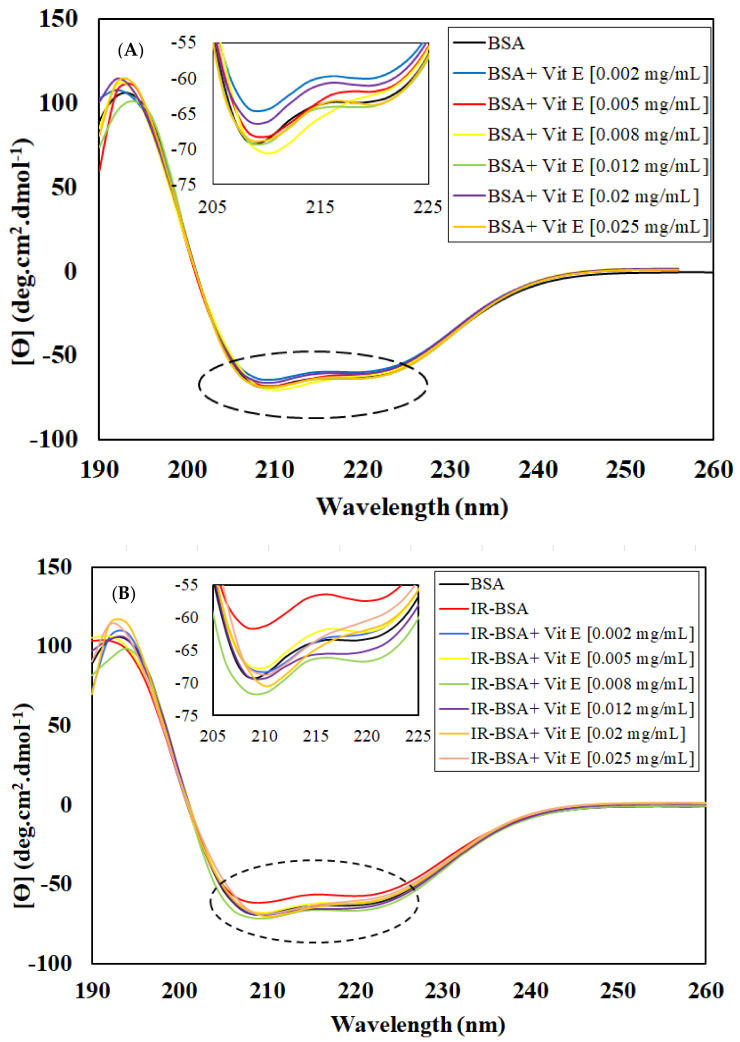
CD spectra in the far-UV region: (**A**) BSA and (**B**) IR-BSA (3 Gy) in the presence of different concentrations of vitamin E. All samples analyzed contained BSA at the concentration of 0.4 mL/mg in 10 mM phosphate buffer (pH = 7) at room temperature.

**Figure 7 antioxidants-10-01875-f007:**
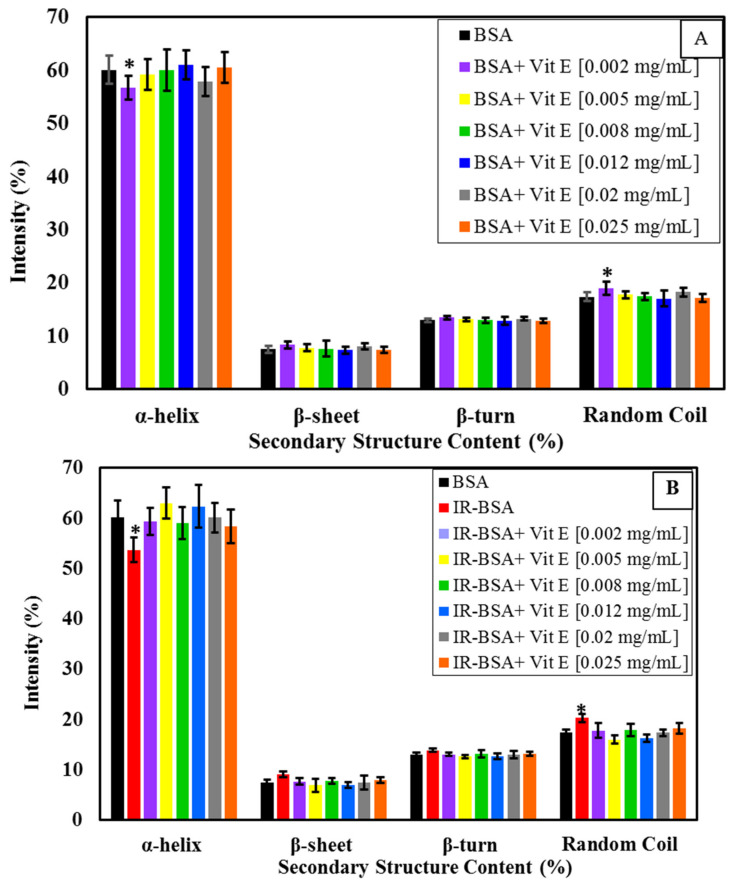
Percentage of secondary structural motifs of BSA (**A**) and IR-BSA (**B**) in the presence of different concentrations of vitamin E obtained from the analysis of the CD spectra using the deconvolution software CDNN2.1. The error bars represent the standard deviation of five measurements. The * sign indicates a *p*-value < 0.05 and a statistically significant difference.

**Figure 8 antioxidants-10-01875-f008:**
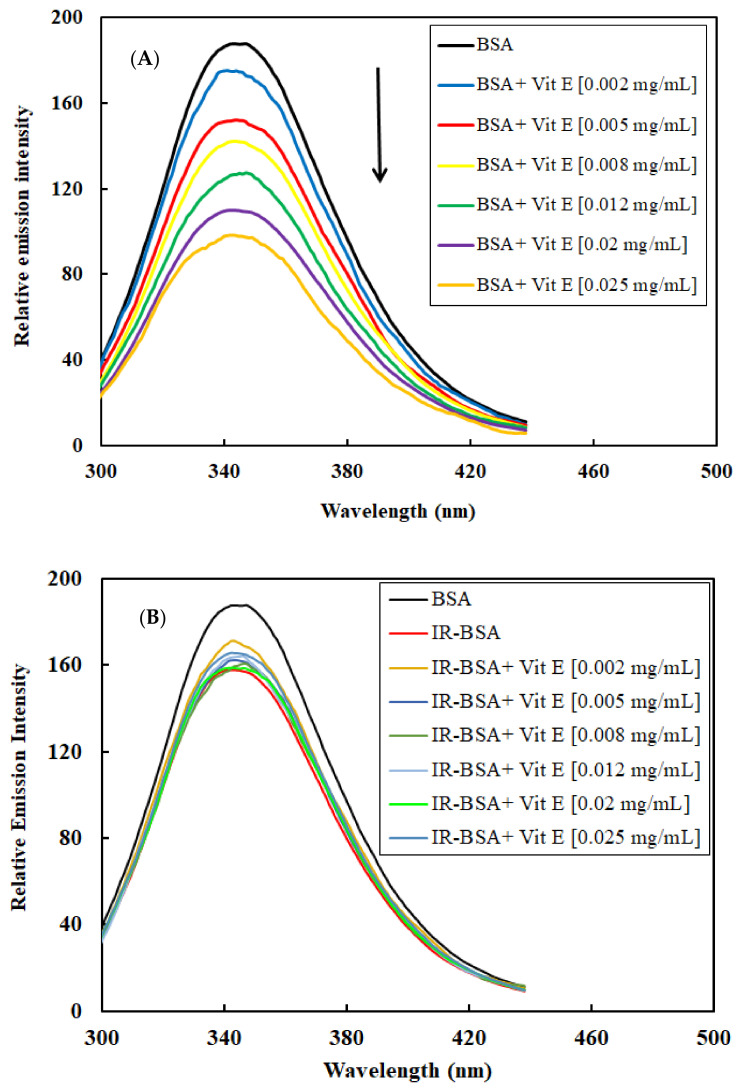
Intrinsic fluorescence emission spectra of BSA (**A**) and IR-BSA (**B**) in the presence of different concentrations of vitamin E. The experimental conditions were the same as mentioned in Figure 6.

**Figure 9 antioxidants-10-01875-f009:**
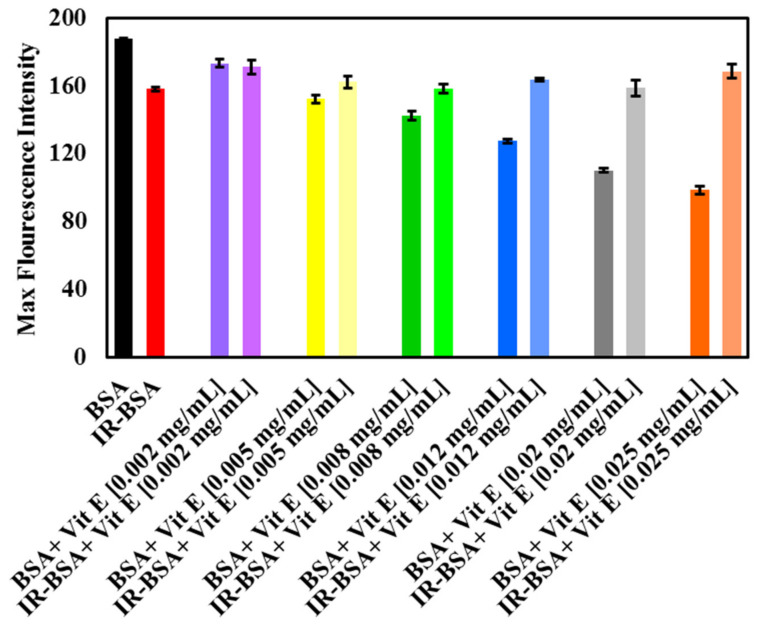
The maximum fluorescence emission of BSA and IR-BSA treated with different concentrations of vitamin E (Vit E). The experimental conditions were the same as mentioned in Figure 8.

**Figure 10 antioxidants-10-01875-f010:**
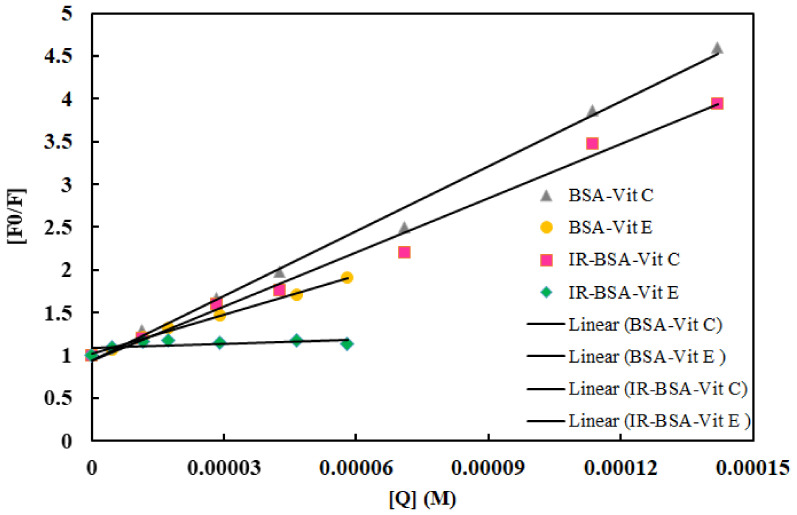
Stern–Volmer plots for binding vitamins C and E to BSA and IR-BSA at room temperature.

**Figure 11 antioxidants-10-01875-f011:**
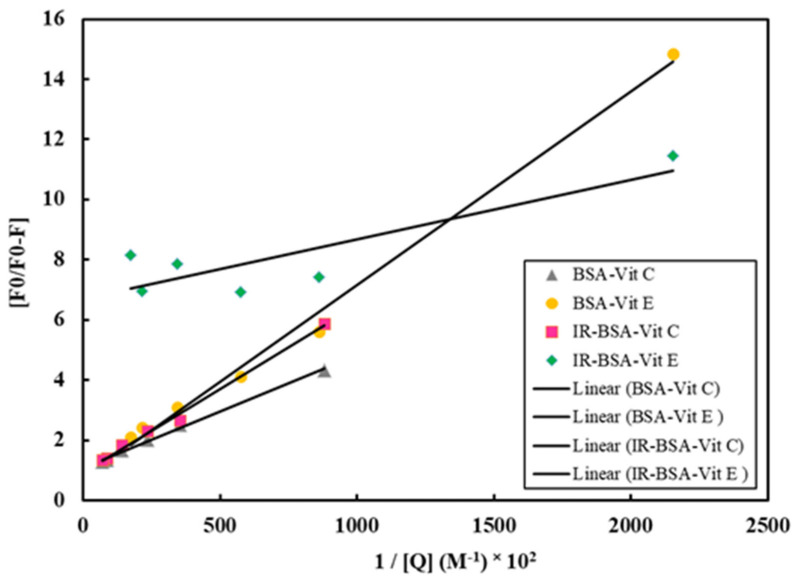
Modified Stern–Volmer plots for the binding of vitamin E to BSA and IR-BSA at room temperature.

**Figure 12 antioxidants-10-01875-f012:**
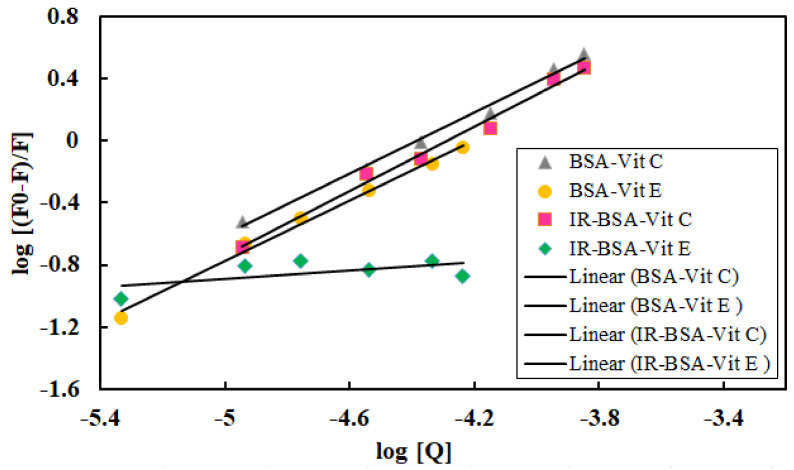
The plots of log (F0 − F)/F versus log[Q] at different concentrations of vitamin C or E.

**Figure 13 antioxidants-10-01875-f013:**
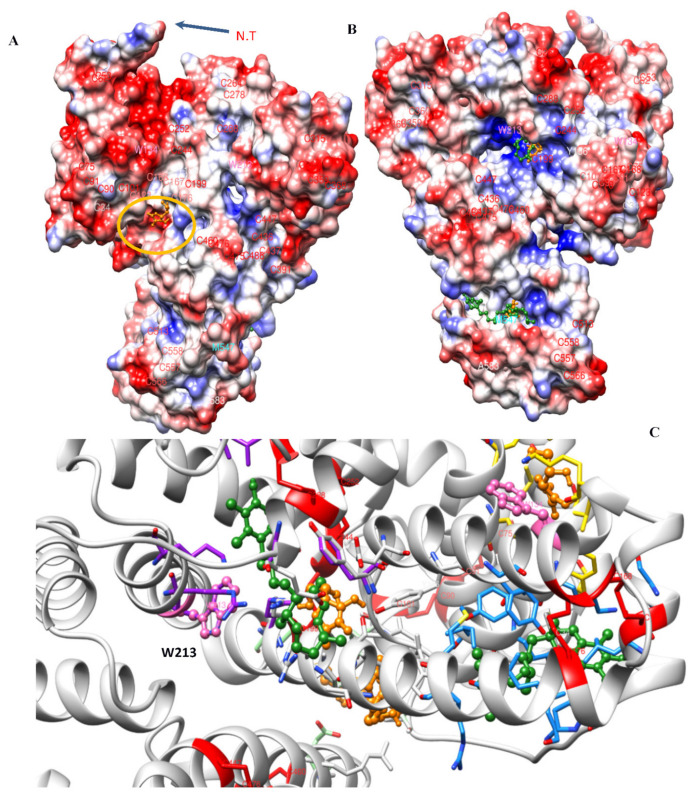
The most probable binding sites of vitamins C (orange) and E (green) on the BSA surface and amino acids involved in these interactions. (**A**,**B**) Electrostatic representations of the protein surface, with negative and positive potentials, are colored in red and blue, respectively. (**C**) Secondary structure illustration of the possible binding sites for vitamins C and E and their proximity to Trp 134 and 213.

**Figure 14 antioxidants-10-01875-f014:**
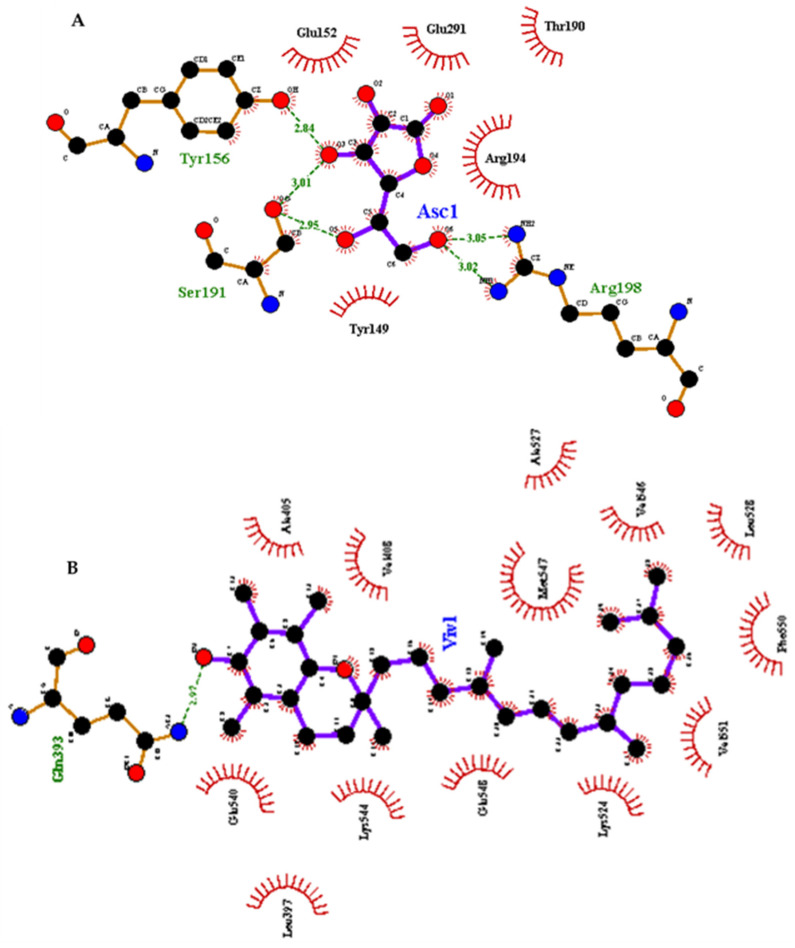
2-D plots of vitamin C (**A**) and vitamin E (**B**) in interaction with BSA selected based on the lowest free energy of interaction.

**Figure 15 antioxidants-10-01875-f015:**
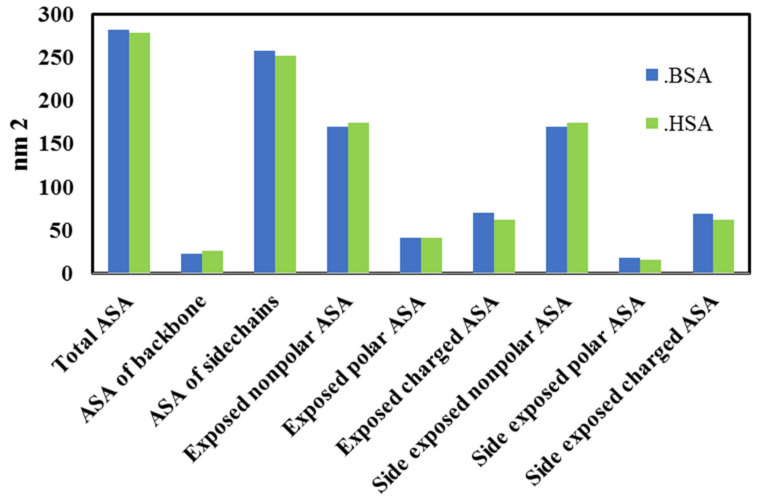
BSA and HSA surface analysis. Accessible surface area is abbreviated as ASA.

**Table 1 antioxidants-10-01875-t001:** Percentage of secondary structural motifs of BSA obtained from the analysis of the CD spectra in Figure 1 using the deconvolution software CDNN2.1.

Dose (Gy)	α-Helix (%)	β-Sheet (%)	β-Turn (%)	Random Coil (%)
0	60.10	7.50	12.98	17.40
0.1	54.73	8.90	13.70	19.83
0.5	54.78	8.88	13.68	19.88
1	54.23	9.03	13.78	20.15
2	54.10	9.08	13.80	20.15
3	53.32	9.25	13.85	20.65

**Table 2 antioxidants-10-01875-t002:** Binding parameters for the different complexes of BSA and IR-BSA with vitamins C (Vit C) and E (Vit-E) at room temperature.

Complex	K_SV_ (M^−1^)	Kq (M^−1^s^−1^)	Ka (M^−1^)	K (M^−1^)	n
BSA–Vit C	25,270	2.527 × 10^12^	27,315	20,792	0.99
IR-BSA–Vit C	21,134	2.1134 × 10^12^	15,593	29,580	1.04
BSA–Vit E	15,241	1.5241 × 10^12^	12,371	12,103	0.97
IR-BSA–Vit E	1592	1.592 × 10^11^	--	1.5	0.1

Stern–Volmer quenching constant (K_SV_), bimolecular quenching rate constant (K_q_), effective quenching constant for the accessible fluorophores (K_a_), binding constant (K), and the number of binding sites (n).

## Data Availability

The data presented in this study are available in the article.

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
