# Peer review of "Radioprotective Role of Vitamins C and E against the Gamma Ray-Induced Damage to the Chemical Structure of Bovine Serum Albumin"

_antioxidants, 2021, doi:10.3390/antiox10121875_

Round 1
Reviewer 1 Report
The authors have addressed most of my concerns. Although I rejected the manuscript before, regarding editor's request, I reviewed the manuscript and it seems significantly improved. However, I will still suggest the introduction section and conclusion should be shorten and concise. The authors should also address this study is for radiotherpay purpose or for radiosafety? I am wondering whether some common free radical scavenger, such as N-acetylcysteine will show the same effect? It may be discussed.
Author Response
Dear Reviewer 1
please see the attachment.
Regards

Reviewer 2 Report
The manuscript brings data regarding the radioprotective effects of Vitamins C and E over BSA molecules against gamma radiation up to 3 Gy. The authors provided a very well-written and scientifically sound manuscript, with a fine experimental design and data analysis to support their conclusions. I have just two suggestions about what I detected throughout the text:
Abstract
The authors cite vitamin C as hydrophobic, when it is actually hydrophilic
Discussion
This part is somewhat confusing. Please review and rewrite it.
Most likely, one α‐Tocopherol radical attacks another, pro- 616 ducing α‐tocopherol dimers, trimmers, and also 4a, 5- epoxy-8a-hydroperoxy α‐tocoph‐ 617 erol that this reaction has a relatively high rate constant (in the order of 103 ) [49]. While α‐ 618 tocopherol dimers and trimers may cause spatial hindering that prohibits its close intrac- 619 tion as well as the interaction of diffusing ROS with BSA.
Author Response
Dear Reviewer 2
please see the attachment.
Regards

This manuscript is a resubmission of an earlier submission. The following is a list of the peer review reports and author responses from that submission.
Round 1
Reviewer 1 Report
The manuscript explores the possible radioprotective effect of the antioxidants ascorbate and tocopherol on a model protein (BSA) by analysis of structure using CD, a fluorescence-based unfolding following gamma irradiation. Due to my level of expertise, I will only comment on the unfolding data (Figs 3 and 5). It is not possible to compare the fluorescence signals between irradiated and non-irradiated for each ascorbate concentration because they are presented in different graphs (Fig 3). The data should be presented so that each ascorbate concentration can be compared. A table would be better than a spectrum and could include error and statistical analysis. The tocopherol effect seems clearer (Fig 5) but should also have its presentation revised. The same comments apply to the comparison of CD data across treatments.
The authors state that the fluorescence signal is due to Trp but it can also arise from Phe. I believe the Phe signal responds most strongly to protein folding/unfolding.
Overall, the English language and presentation can be improved - particularly data analysis, so it is clearer to the reader if there are (statistically) significant effects of the antioxidants. The manuscript would benefit from being shorter/more concisely presented.
Reviewer 2 Report
Zarei et al. submitted the manuscript entitled "Radioprotective Role of C and E Vitamins on the Gamma Rays Induced Damages in the Structure of BSA Protein". There are many flaws in writing, incorrect background information, and data presentation. The authors mixed up the introduction and results, so it is very difficult to understand and follow the logics of authors. They added a prelude before each section of results, but the prelude looks like to explain why they don't do some experiments. They also wrote some information in these paragraphs, but no references were cited in some sentences. The specific comments are listed below:
- There are lots of problems in the introduction, The description of radiation is not precise. The first sentence "The ionization radiation is used in nuclear medical imaging to screen for diagnostic information or treat numerous medical conditions" What nuclear medical imaging can be used by ionizing radiation? How nuclear medical imaging can treat numerous medical condition? What conditions? Please just specify the real problems of radiation, and why the authors or people want to solve these problems. Also, what is gamma radiation? Usually gamma rays or ionizing radiation, and please choose one to present throughout the paper.
- This study focused on the protein structures, including second and tertiary structures would be influenced by radiation. They used Circular dichroism (CD) to conduct experiments and provide many data. Surprisingly, no introduction of this method and the authors did not explain how to decipher the protein structures using CD.
- "ROS activity (oxidative stress) cause deleterious effect on biomolecules causing protein oxidation, misfolding and aggregation, DNA damage, and mutations and lipid peroxidation". This sentence written in the introduction lacks reference.
- Do Vitamin C and Vitamin E were used as natural protector? "Vitamins C and E are two types of natural radiation protectors that have been chosen to investigate their protective effect of BSA structure against the destructive effects of gamma radiation in a dose of 3 Gy." Reference? Vit C and E can scavenge free radicals do not mean they are radioprotector. Please check the definition of radioprotector.
- The percentage changes of alpha-helix and beta-sheets by vitamins are not dramatic, although different. A statistical analysis is required to confirm the significance of these difference.
- The figure 11 should be placed as figure 1 to convince readers about the use of BSA for this study. However, it is not clear how the authors conducted this ASA experiments. No description in the Materials and Methods section. Or this data is from other reference? Please clarify.
- "Figure 9: the most probable binding sites of vitamin C (orange) and E (green) on the BSA surface and amino acids involved in these interactions. A and B)" Although this is a figure legend, please also notice the grammar problem. The first character of the sentence should be capitalized.
- What is the general role (functions) of vitamin C and E on scavenging free radical or eliminating oxidative stress? Does albumin the only protein to be bound by them?
- In section 3.4, "As can be inferred from Figure 9, while the vitamin E is surrounded with hydrophobic side chains the polar chroman head is observable in the positively charged patch close to Trp 213. As it is shown although both vitamins can attach to various part of the protein however the 2D interaction plot for the lowest free energy of binding is represented in Figure 10." There are lots of problems in English writing in these two sentences. Actually, the same problems were shown in other paragraphs of this section.
- In Molecular docking section of Materials and Methods, the software used for analysis should have detailed information , such as version and vendor, or weblink. For example, No these information for the chimera software.
- 3 to 5Gy gamma rays are equal or over therapeutic dose, These dose range does not belong to low dose radiation. Please rephrase it. In the introduction, "In previous studies, the effect of low dose (3 and 5 Gy) gamma radiation on the molecular structure, size distribution and surface charge of BSA was investigated by several spectroscopic methods." There is even no reference for this sentence.